# Dexmedetomidine Has Differential Effects on the Contractility of Equine Jejunal Smooth Muscle Layers In Vitro

**DOI:** 10.3390/ani13061021

**Published:** 2023-03-10

**Authors:** Nicole Verhaar, Susanne Hoppe, Anna Marei Grages, Kathrin Hansen, Stephan Neudeck, Sabine Kästner, Gemma Mazzuoli-Weber

**Affiliations:** 1Clinic for Horses, University of Veterinary Medicine Hannover, 30559 Hannover, Germany; 2Institute for Physiology and Cell Biology, University of Veterinary Medicine Hannover, 30173 Hannover, Germany; 3Small Animal Clinic, University of Veterinary Medicine Hannover, 30559 Hannover, Germany

**Keywords:** intestine, motility, postoperative ileus, ischaemia, reperfusion, organ bath, basal contractile activity, α2 agonist, sedation, colic

## Abstract

**Simple Summary:**

Alpha-2 agonists are commonly used sedatives in horses. They are known for their inhibitive effects on gastrointestinal motility, which limits their use in horses with colic. Dexmedetomidine belongs to the α2 agonist drug class, and studies in human patients have reported that it may enhance gastrointestinal function instead of inhibiting it. Therefore, the aim of this study was to investigate the effect of dexmedetomidine on intestinal smooth muscle function in horses. To evaluate this in varying degrees of intestinal damage, tissue samples were taken from 12 horses prior to and during the disruption of small intestinal blood flow (pre-ischaemia and ischaemia), as well as following the reinstatement of blood supply (reperfusion). We found that the circular smooth muscle (CSM) contractility was not affected by ischaemia, whereas the longitudinal smooth muscle (LSM) showed an increase in both spontaneous and nerve mediated contractile activity. The addition of dexmedetomidine caused a decrease in the spontaneous contractile activity of CSM, but an increase in that of LSM. During ischaemia, dexmedetomidine also mildly increased the nerve mediated contractile activity. These results may indicate a stimulatory effect of dexmedetomidine on small intestinal contractility. However, the influence of dexmedetomidine administration on intestinal motility in vivo needs to be further investigated.

**Abstract:**

α2 agonists are frequently used in horses with colic, even though they have been shown to inhibit gastrointestinal motility. The aim of this study was to determine the effect of dexmedetomidine on small intestinal in vitro contractility during different phases of ischaemia. Experimental segmental jejunal ischaemia was induced in 12 horses under general anaesthesia, and intestinal samples were taken pre-ischaemia and following ischaemia and reperfusion. Spontaneous and electrically evoked contractile activity of the circular and longitudinal smooth muscles were determined in each sample with and without the addition of dexmedetomidine. During a second experiment, tetrodotoxin was added to determine if the effect was neurogenic. We found that the circular smooth muscle (CSM) contractility was not affected by ischaemia, whereas the longitudinal smooth muscle (LSM) showed an increase in both spontaneous and induced contractile activity. The addition of dexmedetomidine caused a decrease in the spontaneous contractile activity of CSM, but an increase in that of LSM, which was not mediated by the enteric nervous system. During ischaemia, dexmedetomidine also mildly increased the electrically induced contractile activity in LSM. These results may indicate a stimulatory effect of dexmedetomidine on small intestinal contractility. However, the influence of dexmedetomidine administration on intestinal motility in vivo needs to be further investigated.

## 1. Introduction

Motility disorders of the small intestine are associated with high mortality rates in horses [1,2,3]. The most common manifestation is a post-operative ileus following colic surgery for strangulating intestinal lesions, with an initial neurogenic and subsequent inflammatory phase inhibiting intestinal motility [4,5]. Many factors can contribute to reduced gastrointestinal (GI) motility, such as intestinal manipulation or anastomosis and the use of opioids or sedatives [6,7,8]. Even though α2-adrenergic agonists can inhibit intestinal motility, the use of these drugs for sedation or analgesia during the perioperative period of colic surgery may be inevitable in some cases. Looking more closely at the evidence available for the inhibition of intestinal motility by α2 agonists, experimental trials in healthy horses have shown that xylazine, detomidine, and romifidine decrease the myoelectrical and/or mechanical activity of the small intestine in an experimental setting [9,10,11,12,13,14]. Other reported GI effects of this drug class are decreased borborygmia following detomidine administration [15] and dose-dependently reduced intestinal smooth muscle contractions caused by xylazine and detomidine addition in vitro [16]. A peripheral α2-adrenoceptor antagonist Vatinoxan was shown to prevent (me)detomidine mediated decrease in borborygmia, indicating that this hypomotility is most likely mediated through activation of the peripheral α2-adrenoceptor [15,17].

Medetomidine and its active D-isomer dexmedetomidine are more selective α2 agonists that are not as commonly used in horses, yet are known produce effective and safe sedation [18,19,20]. These drugs have also been shown to affect intestinal function, with medetomidine decreasing intestinal circular smooth muscle contractility in vitro [16], and decreased borborygmia being documented the first hour after dexmedetomidine administration in horses and donkeys [18,21].

Remarkably, clinical studies in humans have found that dexmedetomidine may enhance intestinal motility peri-operatively or during the continuous sedation of patients in intensive care. These studies reported a variety of improved clinical GI parameters such as increased intestinal sounds and decreased passage time with dexmedetomidine administration compared with treatment with saline as placebo, lidocaine, morphine, and propofol [22,23,24,25,26,27]. Different mechanisms of action have been proposed for this phenomenon, such as the attenuation of inflammation and ischaemic injury or vagal stimulation [28,29,30]; however, there is no direct evidence available to validate these theories. Experimental trials investigating this phenomenon have yielded conflicting results. Studies performed in laboratory animals have reported both inhibitive and stimulating GI effects of dexmedetomidine [29,31,32,33,34], and one study in healthy human individuals found decreased GI passage [35].

With uncertainty surrounding the topic of dexmedetomidine and its effect on intestinal motility, this brings up the question of how dexmedetomidine influences GI motility in horses. If dexmedetomidine could elicit a comparable stimulation of GI function, as seen in humans, this could offer an alternative to the commonly used GI inhibiting α2 agonists. Therefore, the objective of this study was to investigate the effect of dexmedetomidine on the in vitro contractility of equine jejunal smooth muscle during pre-ischaemia, ischaemia, and reperfusion. The second objective was to compare the response of circular smooth muscle (CSM) and longitudinal smooth muscle (LSM), as these play a different role in intestinal motility and may show different responses to the addition of pharmacological substances, as shown in other species [36]. Further objectives were to investigate if the effect is mediated by the enteric nervous system (ENS), and to assess how ischaemic injury affects intestinal contractility. We hypothesized that dexmedetomidine addition would negatively affect contractility, and that this would be more pronounced in CSM. Furthermore, we hypothesized that this would be independent of ENS, and that ischaemic injury of the intestine would further decrease contractility.

## 2. Materials and Methods

### 2.1. Animals

Twelve horses were subjected to experimental segmental jejunal ischaemia under general anaesthesia. The intestinal tissue samples taken during the different ischaemia phases were used for this controlled in vitro experimental trial. Seven mares, two geldings, and three stallions were selected for this terminal study because of severe musculoskeletal issues. The horses were warmbloods with a mean age of 15.9 ± 7.4 years and weight of 547 ± 53 kg. Physical examination, blood cell count, and faecal egg count were performed to assess the general health of the horses prior to the experiment. At least 2 weeks prior to surgery, the horses were stabled at the facilities of the equine clinic of the University of Veterinary Medicine Hannover, with free access to hay and water. On the day of the trial, food but not water was withheld for 6 h prior to anaesthesia.

### 2.2. Surgical Procedure

Following premedication with 5 µg/kg dexmedetomidine (Dexdomitor, Orion Corporation, Espoo, Finland), general anaesthesia was induced (0.05 mg/kg diazepam, Diazedor, WDT eG, Garbsen, Germany; 2.2 mg/kg ketamine, Narketan, Vétoquinol GmbH, Ismaning, Germany). Isoflurane (Isofluran CP, CP-Pharma GmbH, Burgdorf, Germany) in 100% oxygen combined with a continuous rate infusion of dexmedetomidine (5 µg/kg/h) was used to maintain general anaesthesia. Direct blood pressure measurements were performed in the facial artery, and the mean arterial pressure was maintained between 60 and 80 mmHg by administrating lactated Ringer’s solution (RingerLaktat EcobagClick, B. Braun Melsungen AG, Melsungen, Germany) and dobutamine (Dobutamin-ratiopharm 250 mg, Ratiopharm GmbH, Ulm, Germany) to effect. A pre-umbilical median laparotomy was performed in dorsal recumbency. After checking the intestines for pre-existing abnormalities, a jejunal segment located 7 m oral to the ileocecal fold was taken as control (pre-ischaemia sample). Forty-five minutes after induction, segmental ischaemia was induced by occlusion of the mesenteric arteries and veins with umbilical tape in 2 m of aboral jejunum that was located 1 m oral to the ileocaecal fold. In six horses, no flow ischaemia was induced by complete tightening of the ligatures. In the other six horses, low flow ischaemia was implemented with 80% reduction in intestinal blood flow under monitoring with Laser Doppler fluxmetry (O2C, LEA Medizintechnik GmbH, Giessen, Germany). Ischaemia was maintained for 120 min, and prior to removal of the ligatures, another tissue sample was taken from the ischaemic segment (ischaemia sample). Subsequently, reperfusion was initiated, followed by resection of the reperfusion sample after 120 min of reperfusion. The horses were euthanized with 90 mg/kg pentobarbital administered intravenously (Release 50 mg/mL, WDT eG, Garbsen, Germany) after the final sample was taken.

### 2.3. Organ Bath

#### 2.3.1. In Vitro Experiment—Day 1

The jejunal samples were placed in a modified Krebs−Henseleit buffer (117.0 mmol/l NaCl, 4.7 mmol/l KCl, 2.5 mmol/l CaCl_2_, 1.2 mmol/l MgCl_2_, 1.2 mmol/l NaH_2_PO_4_, 11.0 mmol/l Glucose, and 25 mmol/l NaHCO_3_) at 4 °C and transferred to the Institute for Physiology and Cell Biology. For each time point (pre-ischaemia, ischaemia, and reperfusion) from all horses (*n* = 12), eight smooth muscle tissue strips including CSM and LSM of equal size (1 × 0.5 cm) and weight were prepared from each jejunal sample. Consequently, there were two tissue strips used as the control samples and two tissue strips used to test the effect of dexmedetomidine for each type of smooth muscle tested (Figure 1). In the analyses following the experiment, the values of these duplicates were averaged.

The tissue strips were placed into an organ bath filled with 12 mL of modified Krebs−Henseleit buffer (117.0 mmol/l NaCl, 4.7 mmol/l KCl, 2.5 mmol/l CaCl_2_, 1.2 mmol/l MgCl_2_, 1.2 mmol/l NaH_2_PO_4_, 11.0 mmol/l Glucose, and 20 mmol/l NaHCO_3_) with constant aeration with 95% O_2_ and 5% CO_2_, and mounted on an isometric force transducer (Hottinger Baldwin Messtechnik, Darmstadt, Germany). The initial tension of the muscle strips was adjusted to 2 g, which corresponds to 20 mN. Isometric contractile forces of smooth muscle tissues were continuously measured using a chart recorder (4.8 kHz/direct current; Spider 8 chart recorder, Hottinger Baldwin Messtechnik) until the end of the experiment at 120 min, and data were collected with data acquisition software (catmanEasy software, version 1.01, Hottinger Baldwin Mess-technik). The viability and response of the enteric nervous system and smooth muscle cells was tested with electrical field stimulation (EFS) with pulses of 0.5 ms during 10 s with 10 Hz and 30 V at 65, 95, and 110 min (Figure 1). At 80 min (t80), 1 μM dexmedetomidine hydrochloride (Dexdomitor, Orion Corporation) was added to the organ bath chambers. Spontaneous contractile activity was determined based on baseline tension (g), frequency (peak/min), amplitude (mN), and mean active force (MAF; mN) at t75–80 and t105–110 (Figure 1). These time points were selected for the analysis, after visual inspection of all of the results, because this time frame included representative spontaneous intestinal motility before and following dexmedetomidine addition. In the dexmedetomidine-treated LSM samples, these values were also determined at t~85, because the visual inspection of the original traces indicated an intense response in the LSM directly following the addition of dexmededetomidine. Induced contractile activity as a response to EFS was expressed in amplitude (mN), which was quantified at t65 and t110.

#### 2.3.2. In Vitro Experiment—Day 2

The full thickness intestinal tissue of the pre-ischaemia samples was stored in the refrigerator at 4 °C overnight, and on day 2 of the experiment, the tissue strips of smooth muscle were freshly prepared. The aim of this experiment was to test the response to dexmedetomidine with and without tetrodotoxin (TTX) to determine if the changes elicited by dexmedetomidine were nerve dependent. This part of the experiment was only performed with the pre-ischaemia samples of eight horses. The vitality of the tissues was tested by EFS. Then, 1 μM dexmedetomidine was added to all of the samples, with and without 1 μM TTX. The time schedule was comparable to that of the experiment on day 1, with the addition of TTX at t70 and the addition of dexmedetomidine at t80. Successful inhibition was verified by a lack of response to EFS.

### 2.4. Data Analysis

A power analysis was performed before commencing the study (G*Power 3.1.9.1s). To detect a difference of 10 mN in amplitude between the treatment groups with a standard deviation of 5 mN, based on a power of 0.8 and alpha of 0.05 through the use of a two-tailed unpaired two sample *t*-test, a total sample size of 12 horses was required.

Statistical analysis and graph design were performed using commercially available software (Excel 2016, Microsoft, Redmond, WA, USA; Graphpad Prism 9.4.1, Graphpad Software Inc., San Diego, CA, USA). Data were presented as mean (±standard deviation). Data were tested for normal distribution using a Shapiro−Wilk test. The values for basal contractile activity and response to EFS were compared between the groups subjected to low flow and no flow ischaemia using a two-tailed unpaired two sample *t*-test. As there were no significant differences between the horses subjected to the different low-flow and no-flow ischaemia types, the results of both groups were combined for further statistical analysis. Statistical significance was set at *p* < 0.05.

Differences between the different ischaemia phases at t75–80 were compared using a repeated measures ANOVA with Greenhouse–Geisser correction of the *p*-values, followed by a post hoc Tukey test for multiple pairwise comparisons. To evaluate the development over time in the organ bath with or without dexmedetomidine, the basal contractility was compared between t75–80 and t105–110 and the response to EFS between t65 and t110 using a two-tailed paired *t*-test. A comparison between the samples treated with and without dexmedetomidine was performed for each time point using a two-tailed unpaired two sample *t*-test.

## 3. Results

### 3.1. Ischaemia Model

There were no significant differences between the groups subjected to low-flow and no-flow ischaemia. Therefore, the results of the two ischaemia groups were pooled for further data presentation and analysis.

### 3.2. Basal Contractile Activity

#### 3.2.1. Ischaemia Phases

Comparing the different ischaemia phases at t75–80, ischaemia and reperfusion did not affect the contractility of the CSM in any of the tested variables. In contrast, the ischaemic and reperfused LSM showed a significantly higher frequency, amplitude, and MAF compared with the pre-ischaemic values (Figure 2).

#### 3.2.2. Development over Time without the Addition of Dexmed

Comparing the different time points t75–80 and t105–110 of the organ bath experiment, CSM showed an elevated amplitude in the ischaemia samples and an elevated MAF in the reperfusion samples at t105–110 (Figure 3). LSM showed a significant increase in amplitude over time in all of the samples. LSM frequency was elevated in the ischaemia samples only, whereas MAF increased over time in the ischaemia and reperfusion samples.

#### 3.2.3. Effect of Dexmedetomidine Addition

Evaluating the initial response to dexmedetomidine addition at t~85, all of the tested variables of the basal contractile activity were significantly increased in the LSM of all ischaemia phases, whereas no significant initial response of CSM was observed.

Comparing the basal contractility at t105–110 between the samples with and without dexmedetomidine addition, the pre-ischaemia and ischaemia CSM exhibited a significantly lower frequency in the dexmedetomidine-treated samples (Figure 4). Furthermore, the ischaemia CSM showed a significantly lower MAF with dexmedetomidine. Dexmedetomidine did not affect the frequency of LSM. In contrast, LSM amplitude and MAF were significantly higher in the dexmedetomidine-treated samples of all of the ischaemia phases (Figure 4).

### 3.3. Electrical Field Stimulation

#### 3.3.1. Ischaemia Phases

The amplitude of EFS that induced the contractile activity of CSM was comparable during the different ischaemia phases. In LSM, the ischaemia and reperfusion samples exhibited higher amplitudes than the pre-ischaemia samples (Figure 5).

#### 3.3.2. Development over Time without the Addition of Dexmed

CSM and LSM showed a significant change in response to EFS comparing t110 with t65, with an increased amplitude in all of the ischaemia phases (Figure 6).

#### 3.3.3. Effect of Dexmedetomidine Addition

There were no significant differences in response to EFS of CSM for any of the ischaemia phases. In LSM, the amplitude was significantly higher in the dexmedetomidine-treated ischaemia samples (*p* = 0.02) compared with the untreated ischaemia samples (Figure 7). During the other ischaemia phases, this did not reach statistical significance (*p* = 0.059 and *p* = 0.80 for pre-ischaemia and reperfusion, respectively).

### 3.4. Day 2 of the In Vitro Experiment—Dexmedetomidine with and without TTX

All of the tissue samples used in these experiments showed vitality, as they exhibited spontaneous motility and they responded to EFS. Comparing the basal contractile activity at t105–110 of the pre-ischaemia samples that were treated with dexmedetomidine with and without the addition of TTX, there were no significant differences between these groups (Figure 8).

## 4. Discussion

The main finding of the study was that the in vitro addition of dexmedetomidine caused a decrease in the spontaneous contractile activity of CSM, but an increase in that of LSM. The in vitro addition of dexmedetomidine also stimulated the nerve-mediated contractile activity of LSM, yet to a lesser extent. Therefore, we partially rejected the first hypothesis that dexmedetomidine addition would negatively affect contractility. The changes in basal contractile activity were not affected by the addition of TTX, indicating that this was not mediated by ENS, confirming the second hypothesis. The contractility of CSM was not affected by the different phases of ischaemia. In contrast, LSM showed an increased basal and EFS induced contractile activity following ischaemia and reperfusion, leading to the rejection of the third hypothesis.

The stimulating effect of ischaemia on the spontaneous contractile activity of LSM found in the current study is in contrast with studies performed in various species reporting that ischaemia reduced the intestinal contractility of the affected and neighbouring intestinal segments [29,37,38,39,40]. Discrepancies between some of these studies and the current investigation could be attributed to differences between species, in which the SM layer was investigated, or for the variance in ischaemia duration (15 min vs. 2 h). The increase in LSM contractile activity may be caused by a stress response with sympathetic activation leading to initial hypermotility. Circulating catecholamines could affect SM directly, considering that norepinephrine has been shown to elicit excitatory effects in the human colon when binding at α1-adrenoceptor D subtypes [41]. Furthermore, the stress-related activation of intestinal corticotropin-releasing factor (CRF) receptors could mediate the stimulation of motility [42]. Regarding the clinical occurrence of a postoperative ileus in the horse, there is only sparse evidence for the pathophysiologic mechanism in this species [5]. In general, rodent models have shown that sympathetic stimulation can lead to decreased motility in this disease entity [43].

Dexmedetomidine increased the spontaneous and nerve-mediated contractile activity of LSM. This response differed from the effect on the circular layer, where it decreased CSM contractile activity. This corresponds with previous investigations in horses and rats that have shown a varying response to α2-adrenoceptor agonists in the different muscle layers, with inhibited or unchanged CSM contractility, as well as unchanged or enhanced LSM contractility [36,44,45]. These differences may be caused by asymmetric innervation of CSM and LSM with possible variation in α-adrenoreceptor distribution, and may facilitate a reciprocal contractile activity of the two muscle layers [46,47].

In the current study, TTX addition did not change the effect of dexmedetomidine, hence it can be concluded that the observed effect on the spontaneous contractile activity was not mediated through ENS. In line with our results, other studies have also shown that spontaneous contractile activity in equine jejunum and ileum was independent of ENS [16,48]. The effect of dexmedetomidine in this context could be elicited through direct stimulation of receptors on the smooth muscle itself [49]. In the rat ileum, α2 adrenoceptors were shown to be negligible in direct muscular effects of adrenergic agonists. However, this may not be the case for other species, considering the major interspecies differences in α2-adrenoceptors’ subtypes and distribution [45,50]. There are no data available on the distribution of α2-adrenoceptor subtypes in horses, complicating direct comparison and the extrapolation of the results obtained in other species. Apart from direct stimulation of the smooth muscle, other possible explanations for the observed effect on the spontaneous contractile activity may include the activation of adrenoreceptors on the platelet-derived growth factor receptor α+ cells (PDGFRα+ cells), which are electrically coupled with smooth muscle cells in the intestine [41]. Alternatively, it could influence the pacemaker activity of the interstitial cells of Cajal (ICC), which can be independent of ENS [51]. However, the presence of α adrenoceptors on intestinal ICC has not been clarified completely [41,52], and there are no studies available investigating this in horses. Another possible mechanism of action of dexmedetomidine could be an effect on the enteric glia Ca^2+^ response [31], which alters the membrane potential of the smooth muscle cell.

The increased nerve mediated contractile activity may be caused by dexmedetomidine enhanced stimulation of α receptors located on cholinergic neurons of ENS [45,53]. Another possible mechanism of action may be the activation of nonadrenergic imidazoline receptors by dexmedetomidine [54], considering that presynaptic imidazoline receptors have been indicated as a pathway to attenuate ENS response by modulating intestinal cholinergic neurotransmission [54]. However, this could not be confirmed in guinea pig ileum, revealing that imidazoline-like compounds only modulated cholinergic neurotransmission through the interaction with presynaptic α2 adrenoceptors instead of imidazoline receptors [53]. Therefore, this theory is less plausible, yet again, with the limitation of the extrapolation of experimental data between different species.

Reviewing the previous studies investigating dexmedetomidine and other α2 agonists on in vitro contractility, both enhanced and inhibited contractility have been reported. Xylazine, detomidine, and medetomidine reduced spontaneous and/or evoked contractility in horse jejunum, guinea pig duodenum, and mice ileum [16,55,56]. On the other hand, two studies in the rat ileum found that dexmedetomidine increased the amplitude of spontaneous contractions without affecting the frequency [34], and that dexmedetomidine reduced the ischaemia induced inhibition of contractility [29]. Differences between experiment duration, species, and the small intestinal segments that were used, complicate the exact comparison of the studies. Furthermore, different tissue preparations were used, with full thickness intestinal samples as well as combined CSM and LSM layers, and not all of the studies investigated both the CSM and LSM contractile activity.

This study found increased in vitro contractility of the LSM with dexmedetomidine treatment, which conflicts with two in vivo studies that found decreased gut sounds following dexmedetomidine administration in horses [18,21]. This illustrates the fact that in vitro contractile activity does not necessarily translate to an increased motility in vivo. Regarding our results, this difference may be due to CSM playing a more dominant role than LSM in equine small intestinal motility, considering the relative thickness of CSM compared with LSM. Apart from the local effects of dexmedetomidine on the intestinal smooth muscle, its influence on other parts of the intestine or organs may also be of relevance for its ultimate effect on GI motility. The anti-inflammatory properties of dexmedetomidine may reduce the leukocytic inhibition on motility [28,30], and its reduction in ischaemia reperfusion-related injury of the mucosa [57,58] could indirectly ameliorate both the neurogenic and inflammatory inhibition of motility. Moreover, dexmedetomidine can alter the intestinal perfusion, with different authors reporting both improvement and deterioration of intestinal circulation [59,60,61]. Considering that pain is a significant motility inhibiting factor, the analgesic effects of α2 agonists may also be of significance in this matter [13,62]. Finally, there are many differences between the in vitro situation compared to in vivo that influence the extrapolation of the results, such as the migrating myoelectrical complex, luminal filling, intraluminal pressure, and an interaction between the intact muscle layers and the myenteric plexus [46,51,63,64].

A limitation of the study is the use of systemic dexmedetomidine during anaesthesia as premedication and as continuous rate infusion during anaesthesia. This was selected because it is considered unethical to induce general anaesthesia without proper premedication. The continuous rate infusion during the maintenance of anaesthesia was elected as part of the analgesic protocol and as isoflurane sparing anaesthetic management. Consequently, dexmedetomidine was elected to rule out the influence of any other anaesthetics. Any effect would be present in both groups and most likely be temporary, considering that rapid wash-out has been described in organ baths [65]. The measurement of dexmedetomidine levels in the intestinal tissue prior to the organ bath experiment or at the end of the in vitro experiment could have given more information on the relevance of this in vivo dexmedetomidine CRI. The administration of dexmedetomidine prior to ischaemia could have ameliorated the intestinal injury by pharmacological preconditioning, which could have influenced the effect of the ischaemia model [57]. On the other hand, it is unlikely that this affected the comparison of the treatment groups in the contractility measurements, considering this affect would be present in both groups. Furthermore, we confirmed the presence of significant intestinal injury during histological examination of these samples, as published elsewhere [66].

Comparing the in vitro applied concentration of dexmedetomidine with clinical doses, the dexmedetomidine plasma concentrations reached after commonly applied doses of 3.5–5 µ/kg are around 5 ng/mL [18,19], corresponding to approximately 25 nM. This is lower than the concentrations used in the current study, and thus represents another limitation of this work. For this in vitro study, a concentration was elected within the lower range of those previously described in in vitro studies in other species. Even though the investigated drug concentrations are not directly transferable to the clinical situation, the current observations serve as a first step in elucidating the mechanism of dexmedetomidine induced changes on the small intestinal contractility in horses. Further limitations of this study include the small sample size, the individual variation between the horses, and the relatively short reperfusion time.

## 5. Conclusions

The contractile activity of CSM was not affected by ischaemia, yet the LSM contractility increased following ischaemia and reperfusion. Dexmedetomidine inhibited the spontaneous contractile activity of CSM, whereas it stimulated that of LSM. This was not mediated by the enteric neurons, possibly indicating a direct effect on the smooth muscle cells. Dexmedetomidine also mildly increased the electrically induced contractile activity in LSM, which most likely indicates a differential distributionof the α-receptors between the two muscle layers. How the inhibitive effect on CSM and the excitatory effect on the LSM translate to motility in the clinical patient remains unclear, yet CSM may play a more dominant role in peristalsis due to its thickness in relation to that of LSM. Many other factors such as pain, inappetence, or the primary disease may also influence the final effect of dexmedetomidine on the case outcome. To fully comprehend the impact of dexmedetomidine on motility, investigations in perfused ex vivo models or in vivo techniques may be performed in future studies to clarify the many remaining questions. If dexmedetomidine could elicit a comparable stimulation of GI function as seen in man, this could offer an alternative to the commonly used GI inhibiting α2 agonists.

## Figures and Tables

**Figure 1 animals-13-01021-f001:**
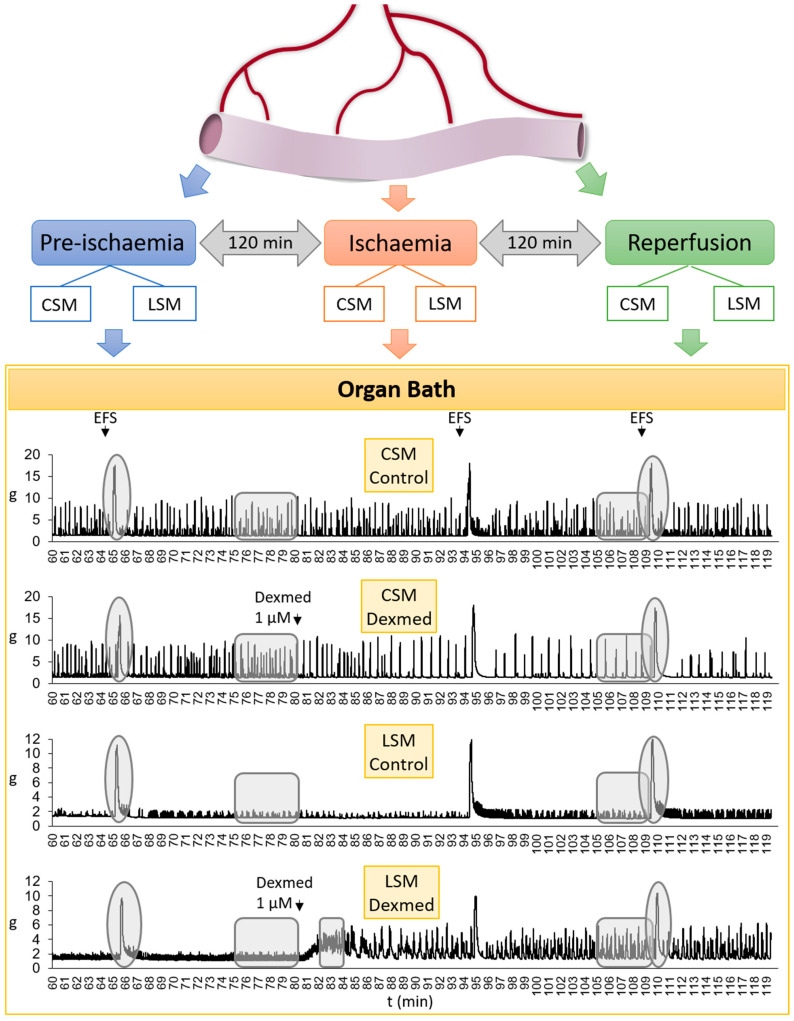
Diagram depicting the timing of the experiment on day 1 including the original traces of representative samples of circular and longitudinally oriented jejunal smooth muscle strips (CSM and LSM, respectively) with and without dexmedetomidine (Dexmed) addition. The oval shaded areas represent the induced contractile activity following electrical field stimulation (EFS) at t65 and t110. The rectangular shaded areas represent the basal contractile activity at t75–80 and t105–110 (additional t~85 for LSM Dexmed).

**Figure 2 animals-13-01021-f002:**
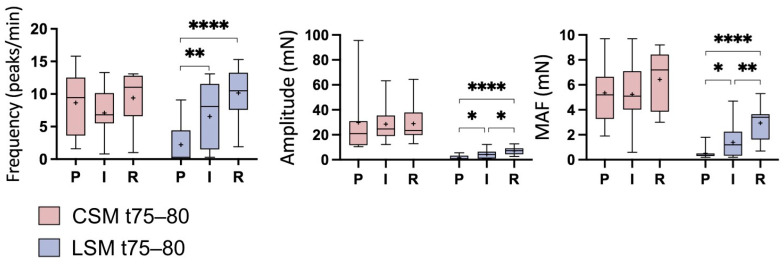
Box plot diagram of the basal contractile activity of jejunal circular smooth muscle (CSM) and longitudinal smooth muscle (LSM) of 12 horses during pre-ischaemia (P), ischaemia (I), and reperfusion (R) at t75–80 in the organ bath. Significant differences between the ischaemia phases (repeated measures ANOVA) are marked with an asterisk (* *p* < 0.05; ** *p* < 0.01; **** *p* < 0.0001). The horizontal bar displays the median and the plus is the mean, the interquartile range is represented by the box and the minimum and maximum by the whisker plots. MAF is the mean active force.

**Figure 3 animals-13-01021-f003:**
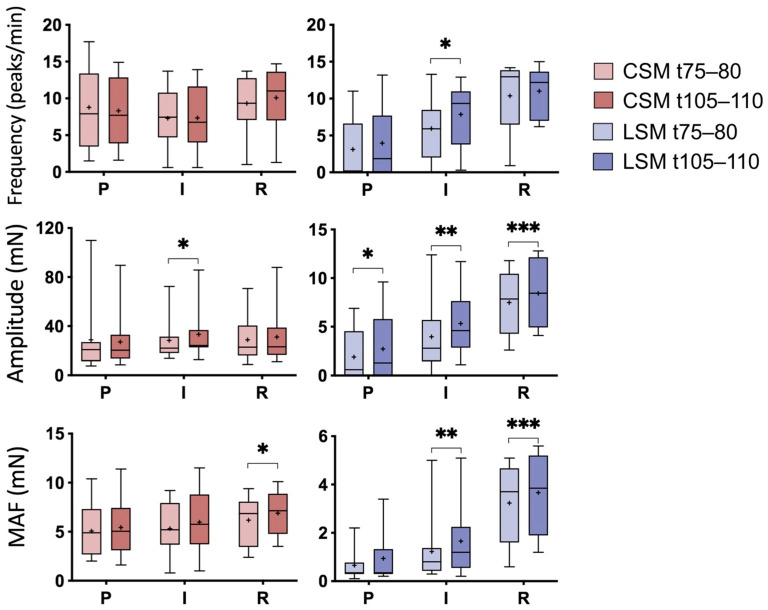
Box plot diagram of the basal contractile activity of jejunal circular smooth muscle (CSM) and longitudinal smooth muscle (LSM) of 12 horses during pre-ischaemia (P), ischaemia (I), and reperfusion (R) at different time points t75–80 and t105–110 during the organ bath experiment. Significant differences between the time points (paired two tailed *t*-test) are marked with an asterisk (* *p* < 0.05; ** *p* < 0.01; *** *p* < 0.001). The horizontal bar displays the median and the plus is the mean, the interquartile range is represented by the box and the minimum and maximum by the whisker plots. MAF is the mean active force.

**Figure 4 animals-13-01021-f004:**
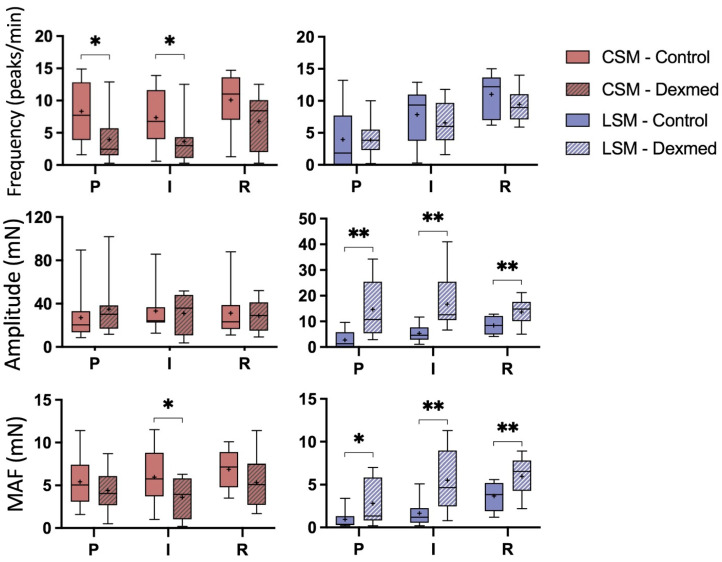
Box plot diagram of the basal contractile activity of jejunal circular smooth muscle (CSM) and longitudinal smooth muscle (LSM) with and without the addition of dexmedetomidine (Dexmed) during pre-ischaemia (P), ischaemia (I), and reperfusion (R) at t105–110. Significant differences between the control group and the Dexmed-treated group (two sampled *t*-test) are marked with an asterisk (* *p* < 0.05; ** *p* < 0.01). The horizontal bar displays the median and the plus is the mean, the interquartile range is represented by the box and the minimum and maximum by the whisker plots. MAF is the mean active force.

**Figure 5 animals-13-01021-f005:**
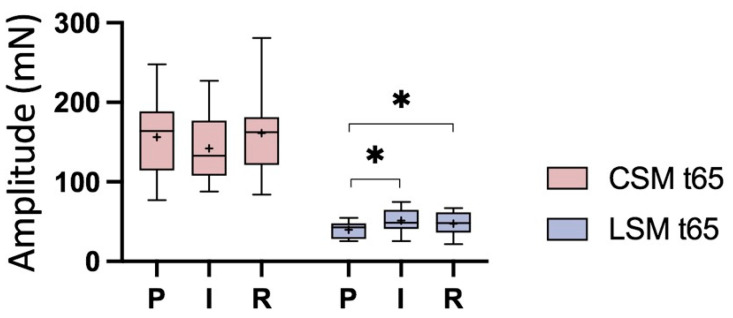
Box plot diagram of the nerve mediated contractile activity of the jejunal circular smooth muscle (CSM) and longitudinal smooth muscle (LSM) of 12 horses during pre-ischaemia (P), ischaemia (I), and reperfusion (R) at t65. Significant differences between the ischaemia phases (repeated measures ANOVA) are marked with an asterisk (* *p* < 0.05). The horizontal bar displays the median and the plus is the mean, the interquartile range is represented by the box and the minimum and maximum by the whisker plots.

**Figure 6 animals-13-01021-f006:**
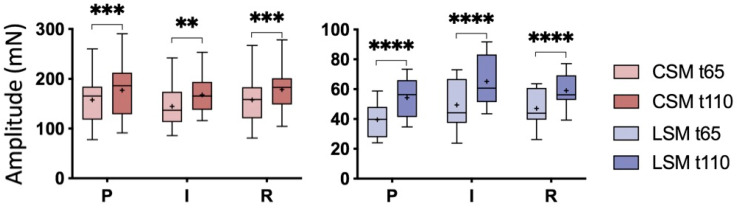
Box plot diagram of the nerve mediated contractile activity of jejunal circular smooth muscle (CSM) and longitudinal smooth muscle (LSM) of 12 horses during pre-ischaemia (P), ischaemia (I), and reperfusion (R) at t65. Significant differences between the time points (paired *t*-test) are marked with an asterisk (** *p* < 0.01; *** *p* < 0.001; **** *p* < 0.0001). The horizontal bar displays the median and the plus is the mean, the interquartile range is represented by the box and the minimum and maximum by the whisker plots.

**Figure 7 animals-13-01021-f007:**
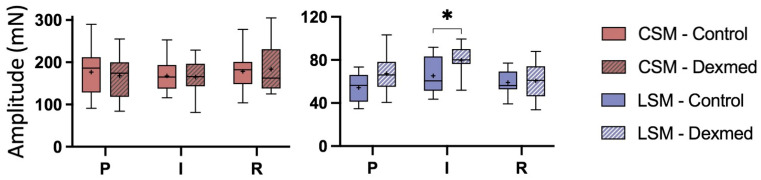
Box plot diagram of the nerve mediated contractile activity of jejunal circular smooth muscle (CSM) and longitudinal smooth muscle (LSM) during pre-ischaemia (P), ischaemia (I), and reperfusion (R) at t110 with and without the addition of dexmedetomidine (Dexmed). Significant differences between the control group and the Dexmed-treated group (two sample *t*-test) are marked with an asterisk (* *p* < 0.05). The horizontal bar displays the median and the plus is the mean, the interquartile range is represented by the box and the minimum and maximum by the whisker plots.

**Figure 8 animals-13-01021-f008:**
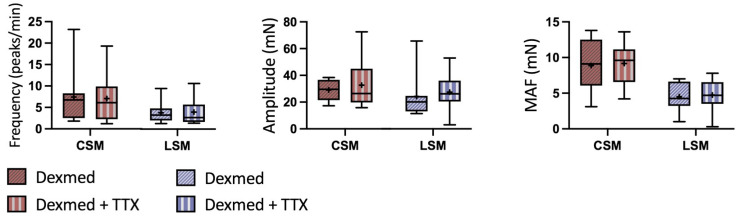
Box plot diagram of the basal contractile activity of jejunal circular smooth muscle (CSM) and longitudinal smooth muscle (LSM) during pre-ischaemia at time point t110 in the organ bath experiment, following treatment with dexmedetomidine (Dexmed), with and without the addition of tetrodotoxin (TTX). There were no significant differences between the treatment groups (two sample *t*-test). The horizontal bar displays the median and the plus is the mean, the interquartile range is represented by the box and the minimum and maximum by the whisker plots. MAF is the mean active force.

## Data Availability

The data presented in this study are openly available in Mendeley Data under the following link http://dx.doi.org/10.17632/3hkh8v9mvv.1 (accessed on 2 March 2023).

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
