# Peer review of "Dexmedetomidine Has Differential Effects on the Contractility of Equine Jejunal Smooth Muscle Layers In Vitro"

_animals, 2023, doi:10.3390/ani13061021_

Round 1
Reviewer 1 Report
in this in vitro study the authors aimed to clarify the effect of dexmedetomidine on intestinal smooth muscle activity.
Since the use of alpha2 agonist is common in equine medicine to treat a variety of condition, mostly thanks to their analgesic effect, it is interesting to highlight its stimulating versus inhibiting effect.
the experimental model is in vitro. As the authors discuss, in vivo results maybe not comparable, so in vivo experimental setting must be set up to confirm what observed.
I only have a couple of suggestions:
- ref 20-42-56, page numbers are missing
- please clarify why those time points are taken into account for the statistical analysis (t85-t110, for example).
Author Response
Dear Reviewer,
Thank you for reviewing our manuscript and for the helpful comments. These are addressed below and in the manuscript.
- ref 20-42-56, page numbers are missing
- Thank you for pointing this out, we corrected this in the reference list.
- please clarify why those time points are taken into account for the statistical analysis (t85-t110, for example).
- This was added to the main text (lines 233-239).
Reviewer 2 Report
Overall, this is an interesting study, and was fairly well designed. I would like it to be better clarified that all segments were exposed to the drug, as this may have affected your results. It was mentioned in the limitations, but there are sections that read as if the controls were not exposed at all, which is inaccurate. The discussion also needs to be better clarified, in regards to species specificity. As different species react differently to drugs, especially in regards to the intestinal tract, if comparisons are made, it should be clear if you are assuming that research in another species is similar when making comparisons. There are also a number of references that are incorrect or could be better sourced. The English language throughout will need to be improved, as there are a number of sentences with incorrect adjectives or awkwardness that sometimes makes it difficult to understand. But overall it is well written.
Line 63: myoelectrical is misspelled.
Line 64: Be sure to examine your references, as #13 measured nociception, not motility.
Line 71: only dexmedetomidine is referenced, an example of medetomidine should also be supplied.
Line 121: as control would sound better
Line 137: at 4oC would be better
Line 138: for each time point would be better
Line 162: was tested with would sound better
Line 172: please expand this section or provide a diagram. It sounds like you only tested bowel from the pre-ischemic samples, but then it appears you tested bowel from the pre, during and post ischemic timepoints. Also, is there any reference to put in the discussion to show that the bowel is still healthy enough to test a day after euthanasia?
Figure 3: it may be advised to consult with a statistician, as it seems illogical that the R samples in the LSM group are truly significantly different as the IQR are almost identical and the means are so close together.
Figure 8: please pick another color for these as I cannot see the means very well, it is too dark.
Line 307: This summary may need to be amended. The surgical procedure was performed with a constant rate infusion of dexmedetomidine, therefore all samples were technically treated with the drug. While it would likely not have an effect in the ex vivo portion directly, as the authors stated there may be other effects on the bowel including inhibition of ischemia and reperfusion, this needs to be discussed as a limitation.
Line 318: comparisons to other species may not be the most valid here. Same for line 331
Line 338: not dependant on or independent of? The sentence is unclear.
Line 342: please expand on this with appropriate references-how do rats differ form horses in regards to the adrenergic agonists? If you are proposing this, the references to support you previous statements must be species specific.
Ref 49: a GI reference is better than a prostate one.
Line 360: thank you for clarifying this is cross species inference, please do so throughout when indicated.
Line 364: contrarily is not a good start to this sentence. It means defiantly or willfully. Please revise. There are a number of awkward sentences throughout the paper, that should be addressed for clarity.
Line 379: representative of
Line 381: as pain or nausea could not have affected your results, this is moot. I would suggest to remove this paragraph, or revise to show ways that intestinal motility could be reduced and contradict your results. The paragraph is not coherent.
Line 400: please provide a more relevant reference. Lidocaine has an extremely short half life, therefore it is not an appropriate comparison.
Line 403: this sentence is confusing-histology was not performed
Line 420: as in vivo cannot be speculated, this sentence should be removed, considering all work thus far contradicts this statement.
Author Response
Dear Reviewer,
Thank you very much for the detailed review of our manuscript. We are grateful for your helpful remarks and also for pointing out the typos in the text. We have addressed your comments in a point-by-point response listed below:
Line 63: myoelectrical is misspelled.
- This was corrected in the main text.
Line 64: Be sure to examine your references, as #13 measured nociception, not motility.
- We agree that the main purpose of this study was nociception. However, in this process the authors also measured duodenal contractions, and stated the following: “a marked, immediate decrease in amplitude of duodenal contractions followed detomidine administration at both doses for 50 minutes.” Therefore we included this reference here as example of mechanical (not myoelectrical) activity. If you still find this reference unsuitable, we are happy to take it out.
Line 71: only dexmedetomidine is referenced, an example of medetomidine should also be supplied.
- We added a reference for medetomidine sedation to the main text (Ref 23, line 72)
Line 121: as control would sound better
- This was changed in the text
Line 137: at 4oC would be better
- This was changed in the text
Line 138: for each time point would be better
- This was changed in the text
Line 162: was tested with would sound better
- This was changed in the text
Line 172: please expand this section or provide a diagram. It sounds like you only tested bowel from the pre-ischemic samples, but then it appears you tested bowel from the pre, during and post ischemic timepoints. Also, is there any reference to put in the discussion to show that the bowel is still healthy enough to test a day after euthanasia?
- It is correct that we only tested the preischemia samples on day 2. We have reworded this section providing a better description of the experimental protocol. Vitality of the tissue was proven showing their spontaneous activity (comparable to the first day) and they all responded to EFS. We also added this important information in the text (line 248 and line 387-8).
Figure 3: it may be advised to consult with a statistician, as it seems illogical that the R samples in the LSM group are truly significantly different as the IQR are almost identical and the means are so close together.
- We agree that this result looks as if it could not be significant, but in the interpretation one should consider that these are two time points that are compared using a paired t-test. Therefore, the difference between the time points for each individual horse are compared. As an example, the table below shows the mean active force (MAF) results for the reperfusion samples. All horses showed an increase over time, and therefore the paired t-test yielded statistical significance (calculated with both excel and graphpad). Another graph type such as connected individual value plots may have presented this information more clearly, yet due to the large amount of data being presented in the manuscript made this graph unsuitable regarding lay out and overview.
|
Horse |
T75-80 |
T105-110 |
|
1 |
0,6 |
1,2 |
|
2 |
4,7 |
5,6 |
|
3 |
4,2 |
4,9 |
|
4 |
3,8 |
3,9 |
|
5 |
1,3 |
1,7 |
|
6 |
5,1 |
5,3 |
|
7 |
2,2 |
2,5 |
|
8 |
1,4 |
1,5 |
|
9 |
4,6 |
4,9 |
|
10 |
4,7 |
5,6 |
|
11 |
3,6 |
3,8 |
|
12 |
2,6 |
3,1 |
Figure 8: please pick another color for these as I cannot see the means very well, it is too dark.
- This figure was changed accordingly
Line 307: This summary may need to be amended. The surgical procedure was performed with a constant rate infusion of dexmedetomidine, therefore all samples were technically treated with the drug. While it would likely not have an effect in the ex vivo portion directly, as the authors stated there may be other effects on the bowel including inhibition of ischemia and reperfusion, this needs to be discussed as a limitation.
- We have altered these sentences (Line 316-323), so that it is more clear that we are reporting the effect of in vitro addition of dexmedetomidine. Furthermore, we elaborated on this topic in the limitations sections (line 422 – 428).
Line 318: comparisons to other species may not be the most valid here. Same for line 331
- We agree that it is difficult to compare results of experiments performed in different species with extremely different gastrointestinal physiology, especially with varying experimental protocols. We reformulated the discussion accordingly, adding the species in which the results were obtained.
Line 338: not dependant on or independent of? The sentence is unclear.
- This sentence was altered accordingly
Line 342: please expand on this with appropriate references-how do rats differ form horses in regards to the adrenergic agonists? If you are proposing this, the references to support you previous statements must be species specific.
- This is not what we intested to convey with this sentence: to our knowledge, it is not possible to discuss what the differences between alpha adroreceptor types and distribution in rats and horses are, because there is no data available for horses. We have tried to clarify this more in the text so that this statement cannot be misinterpreted (Line 354 – 360).
Ref 49: a GI reference is better than a prostate one.
- Good point, we removed this reference and left the two GI references.
Line 360: thank you for clarifying this is cross species inference, please do so throughout when indicated.
- We have tried to add more species information or point out cross species inference throughout the discussion (line 325, 328, 337, 342, 354-358, 377)
Line 364: contrarily is not a good start to this sentence. It means defiantly or willfully. Please revise. There are a number of awkward sentences throughout the paper, that should be addressed for clarity.
- This was meant in that way: the sentence focuses on studies contradicting the ones in the previous sentence. We replaced this with one of its synonyms ‘on the other hand’.
Line 379: representative of
- This was changed in the main text.
Line 381: as pain or nausea could not have affected your results, this is moot. I would suggest to remove this paragraph, or revise to show ways that intestinal motility could be reduced and contradict your results. The paragraph is not coherent.
- The paragraph was altered accordingly to make it more understandable (396 – 405). The paragraph was meant to discuss the difficulties of extrapolating the in vitro results to an in vivo situation. If this is deemed unnecessary then we are willing to shorten this or take the paragraph out.
Line 400: please provide a more relevant reference. Lidocaine has an extremely short half life, therefore it is not an appropriate comparison.
- The reference was replaced with another reference (Line 417). In this study, the recovery times of tissue contractility after washing was tested not only for lidocaine but also for mexiletine, bupivacaine, tetracaine (which all have different half lifes) in equine jejunum.
Line 403: this sentence is confusing-histology was not performed
- This sentence was deleted. Histology was performed in another part of this study, published elsewhere (Grages et al. 2022). This is now mentioned briefly in the discussion (line 424-426).
Line 420: as in vivo cannot be speculated, this sentence should be removed, considering all work thus far contradicts this statement.
- We deleted this sentence.
Reviewer 3 Report
The authors present an interesting in vitro study on the effects of dexmedetomidine on circular and longitudinal smooth muscle contractility in a pre-ischemia, ischemia and post-ischemia model. In order to address the relevance to post-operative ileus (POI) following colic surgery, I suggest the following:
Introduction:
Line 57: Is there any evidence to support which muscle layer(s) are affected in POI? If so, please add.
Line 74: Were the circular and longitudinal muscle laters evaluated separately in reference [16]?
Line 93: Please provide rationale for why you decided to investigate circular and longitudinal muscle layers independently.
Discussion
Line 316: Can you comment on how this relates to POI in vivo? Does sympathetic stimulation lead to non-progressive motility in this disease process?
Line 405: Can you clarify if histology was performed in this study? If so, please report in the methods and results section.
Author Response
Dear Reviewer,
Thank you for reviewing our manuscript and for your useful suggestions. We addressed these in the manuscript and in the point-by-point response below:
Introduction:
Line 57: Is there any evidence to support which muscle layer(s) are affected in POI? If so, please add.
- The studies investigating the functional aspect seem to focus on the CSM, but there is evidence for the significant involvement of the LSM in the inflammation. We incorporated this in the text (line 52-53).
Line 74: Were the circular and longitudinal muscle laters evaluated separately in reference [16]?
- Only the CSM was investigated in this study. This was added to the text.
Line 93: Please provide rationale for why you decided to investigate circular and longitudinal muscle layers independently.
- This was added to the introduction (Line 95-96).
Discussion
Line 316: Can you comment on how this relates to POI in vivo? Does sympathetic stimulation lead to non-progressive motility in this disease process?
- This was added to the end of this paragraph (line 335-338)
Line 405: Can you clarify if histology was performed in this study? If so, please report in the methods and results section.
- This sentence was removed. We addressed the histology further along in the limitations: this was performed in these samples and published elsewhere (line 424-426).
Reviewer 4 Report
I revised the manuscript entitled “Dexmedetomidine has a differential effect on circular and longitudinal intestinal smooth muscle contractility of the equine jejunum in vitro “, which I read with a lot of pleasure. The results of the study are interesting and well structured, its topic is of interest to those who are working in the field. The article is well structured, and in my opinion is suitable for publication after major revision.
In the attached file my comments and suggestions

Author Response
Dear Reviewer,
Thank you for reviewing our manuscript. We appreciate your kind feedback and useful comments that we addressed directly in the manuscript.
The only point that we did not incorporate as suggested, were the references in the introduction, as we did not think these portrayed the mortality following clinical postoperative ileus. We hope you do not mind us adding another recent reference.
Round 2
Reviewer 2 Report
Overall, the manuscript is much improved. There are a few outstanding issues that will need to be addressed for accuracy and to better report your study. One issue, is that there still is no testable hypothesis in the introduction. This must be added, in addition to the objectives, for a study to be complete. Review the references again as well, the discussion is much improved, but there are still shortcomings in the intro. Finally, the graphs are still difficult to read-check them all for visibility and consider reporting the data as means/SD in a bar graph since this is how you compared your results. The medians/boxplots may be a reason the data seems to not accurately show the differences/P values.
Line 52: I suggest limiting references to those that are specifically referencing horses, as the sentences is focused on equine colic surgery. There are a number of studies that have investigated this, and so I tis not necessary to turn to rat or human literature to prove this point (ie Cook, Pihl).
Line 54: same for this sentence. As the ischemia reperfusion model is well studied in the horse, there are enough references in the literature to cite without turning to other species. While many of these papers were published in the 1980-1990, they are still relevant. Please reexamine the introduction and try to limit references to equine papers.
Line 91: please say humans instead of man.
References: there is an extra reference not numbered in the list: Abass…between 22 and 23
Line 92: While the study now has objectives, there still is no testable hypothesis. What is the null and alternative?
Line 181: please provide the temperature of the refrigerator.
Line 220: as in the previous review, contrarily is not appropriate to start this sentence. “ In contrast” is probably what you mean. Same for line 253.
Figure 4: I still am having significant trouble seeing the median lines in the dark red images and even the dark blue. Please pick colors or patterns that clearly show your median lines. Same for figure 6 and 7 and 8. Check the others too-it should be easy to read the figures without blowing them up.
Figure 6: The legend does not define the difference between 2,3 and 4 asterisks. Does it mean anything?
Figure 8 should be referenced in the text as well.
Line 197: your analysis analyzed means not medians- it would be more accurate to include the mean as a dot to show what this was on the box plot, or use a bar graph with error bars, which would be more correct and better represent the data and your P values.
Line 327: I appreciate that you are now contrasting with other species, but two of the references (41,42, are equine-did your study concur or contrast with them as well? Please be more careful with citations.
Line 337: reference 47 and 6 are the same.
Line 391: change “that does not harmonise” to “that conflicts with”
Line 398: not sure what netto means. Replace with net? Or revise to make the sentence more clear.
I am not aware of the journal reference format, but as the references are variable in their presentation, it would be advised to pay careful attention to the format required.
Author Response
Thank your for your review and comments. Below we have added the point-by-point response:
One issue, is that there still is no testable hypothesis in the introduction. This must be added, in addition to the objectives, for a study to be complete.
- This was added to the main text (line 101 -104)
Review the references again as well, the discussion is much improved, but there are still shortcomings in the intro.
- We have taken the references from other species out, except for the paragraph describing the effect of dexmed in humans.
Finally, the graphs are still difficult to read-check them all for visibility and consider reporting the data as means/SD in a bar graph since this is how you compared your results. The medians/boxplots may be a reason the data seems to not accurately show the differences/P values.
- We have changed the graph colors and added dots indicating the median.
Line 52: I suggest limiting references to those that are specifically referencing horses, as the sentences is focused on equine colic surgery. There are a number of studies that have investigated this, and so I tis not necessary to turn to rat or human literature to prove this point (ie Cook, Pihl).
- We agree that there are many equine studies investigating intestinal IR injury in experimental models describing inflammation. However, these are not necessarily focused on pathophysiology of postoperative ileus and contractility measurements, so we did not include these in the manuscript. To our knowledge, the only studies directly relating intestinal inflammation to reduced contractility were performed in rats. Nevertheless, we took these references out to follow your suggestions.
Line 54: same for this sentence. As the ischemia reperfusion model is well studied in the horse, there are enough references in the literature to cite without turning to other species. While many of these papers were published in the 1980-1990, they are still relevant. Please reexamine the introduction and try to limit references to equine papers.
- We have taken the other species references out (see remark above).
Line 91: please say humans instead of man.
- This was changed in the text
References: there is an extra reference not numbered in the list: Abass…between 22 and 23
- I cannot see that in my version, but this may have something to do with the referencing program and/or the mdpi edit. Thank you for pointing this out, I will be sure to check this again after removing the hyperlinks of the referencing program.
Line 92: While the study now has objectives, there still is no testable hypothesis. What is the null and alternative?
- We added the hypotheses to the introduction and discussion (Line 324-329). In order for the introduction to remain readable, we only described the null hypotheses.
Line 181: please provide the temperature of the refrigerator.
- This was added to the text (line 184)
Line 220: as in the previous review, contrarily is not appropriate to start this sentence. “ In contrast” is probably what you mean. Same for line 253.
- This was altered in the text
Figure 4: I still am having significant trouble seeing the median lines in the dark red images and even the dark blue. Please pick colors or patterns that clearly show your median lines. Same for figure 6 and 7 and 8. Check the others too-it should be easy to read the figures without blowing them up.
- The figures were all changed accordingly
Figure 6: The legend does not define the difference between 2,3 and 4 asterisks. Does it mean anything?
- This was left out by mistake, thank you for pointing this out. We added this to the text.
Figure 8 should be referenced in the text as well.
- This was added to the results section.
Line 197: your analysis analyzed means not medians- it would be more accurate to include the mean as a dot to show what this was on the box plot, or use a bar graph with error bars, which would be more correct and better represent the data and your P values.
- We have added the suggested point as mean.
Line 327: I appreciate that you are now contrasting with other species, but two of the references (41,42, are equine-did your study concur or contrast with them as well? Please be more careful with citations.
- Yes, that is what we tried to say, but maybe this was not stated clearly enough: these studies all reported that ischaemia reduced intestinal contractility, regardless of the species. We reworded this to clarify this more (line 333-335).
Line 337: reference 47 and 6 are the same.
- As with your previous remark, I somehow cannot see this in my version. I will look out for this in future versions without the hyperlinks.
Line 391: change “that does not harmonise” to “that conflicts with”
- This was changed in the text.
Line 398: not sure what netto means. Replace with net? Or revise to make the sentence more clear.
- We exchanged netto for ultimate.
I am not aware of the journal reference format, but as the references are variable in their presentation, it would be advised to pay careful attention to the format required.
- Thank you for pointing this out. We will be sure to carefully check this once the reference program hyperlinks are removed.
Reviewer 4 Report
Dear authors, the paper improved a lot, however some points were not addressed adequately in the first revision round, please see the attached file

Author Response
Thank you for your review and comments. Below we have added our point-by-point response:
Line 51 – here I suggest to add that the amount of starch in the diet is one of the main cause of the motility disorders and cite: 10.1186/s12917-022-03289-2 and 10.1186/S12917-022-03433-Y/TABLES/4
- Considering our research is more focused on postoperative ileus and motility disorders following IRI, we do not believe these references necessarily belong here. We suggest that the editor can decide on the relevance of these papers for the introduction of this paper.
Line 435 – I suggest to add the limitations of the study
- Further limitations were added to the text (line 441-443).
Lin 436 – practical implication of the obtained results are missing
- We have attempted to describe this in line 458 – 466. We believe that further conclusions on the practical implications cannot be made based on our in-vitro study, considering the difficulties in translating these results to the in vivo situation, as discussed in the manuscript.
Round 3
Reviewer 4 Report
i suggest to accept the paper